# Glucagon-like Peptide-1 Receptor Agonist Use and Pancreatic Cancer Risk in Patients with Chronic Pancreatitis

**DOI:** 10.3390/cancers18020179

**Published:** 2026-01-06

**Authors:** Sarina Ailawadi, Jennifer E. Murphy, Michael H. Storandt, Amit Mahipal

**Affiliations:** 1Department of Medicine, University Hospitals, Case Western Reserve University, Cleveland, OH 44106, USA; sarina.ailawadi@uhhospitals.org; 2Department of Clinical Research Development, University Hospitals, Cleveland, OH 44106, USA; jennifer.murphy@uhhospitals.org; 3Department of Oncology, Mayo Clinic, Rochester, MN 55905, USA; 4Department of Oncology, University Hospitals Seidman Cancer Center, Case Western Reserve University, 11100 Euclid Ave., Cleveland, OH 44106, USA

**Keywords:** pancreatic cancer, GLP-1 receptor, pancreatitis

## Abstract

Glucagon-like peptide-1 receptor agonists (GLP-1 RAs) are increasingly being prescribed for treatment of type 2 diabetes mellitus (T2DM) and obesity. However, the risk of developing pancreatic cancer with the administration of GLP-1 RAs is currently unclear. In this study, using large healthcare database, we evaluate the association between incidence of pancreatic cancer in 89,596 patients with chronic pancreatitis and intake of GLP1 RAs. GLP-1 RA use was associated with a significantly reduced 5-year incidence of pancreatic cancer in all patients with chronic pancreatitis (HR 0.49, 95% CI 0.30–0.80), as well as the subpopulation with both chronic pancreatitis and T2DM (HR 0.53, 95% CI 0.31–0.91). Future larger studies with longer follow up are needed to confirm the potential beneficial effect of GLP-1 RA on incidence of pancreatic cancer.

## 1. Introduction

Glucagon-like peptide-1 receptor agonists (GLP-1 RAs) are increasingly being utilized for treatment of type 2 diabetes mellitus (T2DM) and obesity [1]. This class of medication mimics endogenous GLP-1 and stimulates the GLP-1 receptor, enhancing insulin secretion from the pancreatic islet cells, inhibiting glucagon release, and slowing gastric emptying, thus improving glycemic control, reducing post-prandial glucose spikes, enhancing satiety, and promoting weight loss [2,3,4]. GLP-1 RA medication prescriptions have risen in the last decade, paralleling the global increase in diabetes and obesity; more than 38 million adults in the United States have T2DM and over 40% of adults meet the criteria for obesity thus making GLP-1 RA’s on of the fastest growing therapeutic classes [5]. There has also been emerging evidence linking cardiovascular risk reduction with GLP-1 RA use in patients who are overweight or who have obesity [6,7].

Despite their benefits, adverse effects of GLP-1 RAs can include nausea, diarrhea, vomiting, constipation, gastroparesis, and pancreatitis [8]. Some studies have demonstrated association between GLP-1 RA use and various malignancies. GLP-1 RAs have been linked to an increased risk of medullary thyroid carcinoma in patients with multiple endocrine neoplasia [9]. Conversely, emerging evidence suggests that GLP-1 RAs may reduce the risk of certain obesity-related cancers, including colorectal and endometrial cancer [10]. However, large randomized trials and observational studies have not consistently demonstrated a clear association with pancreatic cancer [1,10,11]. Therefore, concerns remain regarding the safety of GLP-1 RA use and the risk of pancreatic cancer [12].

Chronic pancreatitis (CP) is a known risk factor of pancreatic cancer, with incidence increasing over time from diagnosis; the risk after 10 and 20 years following a CP diagnosis is 1.8% and 4%, respectively [10,13]. CP is defined as progressive inflammation and irreversible structural changes in the pancreas [13,14]. The persistence of inflammation and tissue injury creates an environment that promotes genetic and epigenetic mutations in the pancreatic islet cells, thus predisposing individuals to precursor lesions and ultimately, pancreatic cancer [14,15]. Patients with chronic pancreatitis have up to a 16-fold increased risk of developing pancreatic cancer [14,15].

Although CP can induce endocrine insufficiency such as diabetes, T2DM has been established as an independent risk factor of pancreatic cancer, with a 6–8-fold-higher risk of developing pancreatic cancer as compared to the general population [14,15]. A possible pathogenetic mechanism includes adipokines which can contribute to oncogenesis by stimulating chronic inflammatory mediators, increasing oxidative stress, and impairing the immune response [1,16].

Patients with CP and T2DM are considered a high-risk population, and when both conditions are coexisting, they can have a multiplicative effect on the risk of developing pancreatic cancer [17]. While case reports and pharmacovigilance data have linked GLP-1RAs to pancreatitis, large randomized trials and observational studies have not consistently shown an association with pancreatic malignancy [12]. Given the widespread use of GLP-1 RAs in T2DM management, it is important to assess whether GLP-1 RA use impacts the incidence of pancreatic cancer in high-risk populations that are at baseline predisposed to this malignancy. To address this gap, we aimed to investigate the association between GLP-1 RA use and the incidence of pancreatic cancer among patients with CP. In addition, we conducted a subgroup analysis in patients with a history of both CP and T2DM.

## 2. Methods

### 2.1. Study Design and Patient Identification

We performed a retrospective study using a de-identified federated database with 111 contributing health care organizations (HCO), including over 146 million patients (TriNetX Research Network, Cambridge, MA, USA (https://www.trinetx.com); date of data access: 15 December 2025). Adult patients diagnosed with chronic pancreatitis (ICD-10: K86.1) from 1 January 2015, onward were identified and grouped based on their subsequent exposure to one of the following GLP-1 RAs: semaglutide, dulaglutide, tirzepatide, exenatide, liraglutide, lixisenatide, and albiglutide. These cohorts were further stratified according to the presence of T2DM (ICD-10: E11).

Patients were excluded if they had prior exposure to any of the GLP-1 RAs, a prior diagnosis of pancreatic cancer (ICD-10: C25), a history of Whipple procedure, or a history of bariatric surgery. To ensure adequate follow-up and accurate cohort stratification, included patients were required to have at least two documented instances of chronic pancreatitis (From 1 January 2015) and at least one follow-up visit after the index date. The index date was defined as the GLP-1 RA initiation date for the exposed cohort and the first recorded diagnosis of chronic pancreatitis for the unexposed control cohort. We acknowledge that this definition results in a longer potential observation period for pancreatic cancer in the control group; however, this approach was necessary to ensure a medication-naïve comparator and allow for an appropriate comparison between GLP-1–exposed and unexposed patients.

### 2.2. Statistical Analysis

To address potential confounding, cohorts were balanced using 1:1 greedy nearest-neighbor propensity score matching based on age at index, sex, race, ethnicity, BMI ≥ 30, HbA1c ≥ 8, and comorbidities including tobacco use, alcohol use, hypertension, diabetes mellitus, hyperlipidemia, obesity, presence of pancreatic cysts, and use of anti-diabetic medications (insulin, metformin, and SGLT-2 inhibitors). For categorical laboratory covariates (e.g., BMI ≥ 30 and HbA1c ≥ 8), TriNetX models each specified category as a binary indicator; patients without available laboratory values are treated as not meeting the category. Continuous variables are reported as means ± standard deviations (SD) and compared using independent *t*-tests. Categorical variables are presented as counts or percentages, with associations tested using chi-squared tests. Covariate balance after propensity score matching was assessed using standardized mean differences, with <0.1 indicating adequate balance; although balance improved across all covariates, some remained above this threshold (Table 1 and Table 2).

The primary outcome was the incidence of pancreatic cancer, assessed from 6 months to 5 years after the index date. Kaplan–Meier curves were generated to evaluate time to pancreatic cancer, and hazard ratios (HRs) with 95% confidence intervals (CIs) were estimated. To quantify the association between GLP-1 exposure and incidence of pancreatic cancer, odds ratios (ORs) with 95% CIs were also calculated. Follow-up time was summarized for each cohort. Any patient that developed pancreatic cancer before the 6-month window was removed from the analysis. A sensitivity analysis using the same statistical methods was conducted exclusively among patients with type 2 diabetes. All analyses were performed using TriNetX online analytics platform.

## 3. Results

### 3.1. Patient Characteristics Before and After Matching

Among 89,596 adult patients with CP, there were 3183 patients who were prescribed a GLP-1 RA (Table 1, Figure 1). Compared to patients who did not receive a GLP-1 RA, those who did were more often older (median age 57.3 vs. 54.6 years), female (51.9% vs. 46.4%), having a BMI ≥ 30 (69.6% vs. 27.9%), having a HbA1c ≥ 8 (56% vs. 11.6%), and having a history of pancreatic cysts (18.5% vs. 12.1%), hypertension (83% vs. 52.5%), obesity (63.1% vs. 16.3%), or tobacco use (15.7% vs. 9.2%, *p* < 0.001 for all, Table 1). Additionally, a greater proportion of patients who received a GLP-1 RA had received insulin (71% vs. 27.3%), metformin (62.3% vs. 11.8%), or an SGLT2 inhibitor (29% vs. 2.3%, *p* < 0.001 for all, Table 1).

After PSM, 3162 patients were included in the GLP-1 RA recipient group and 3162 were included in the group who did not receive a GLP-1 RA (Figure 1). After matching, all variables were balanced in the GLP-1 RA and no GLP-1 RA groups, with the exception of average BMI (33.26 vs. 31.11, *p* < 0.001), BMI ≥ 30 (69.4% vs. 72%, *p* < 0.05), hypertension (82.9% vs. 84.9%, *p* < 0.05), obesity (62.9% vs. 66.5%, *p* < 0.01), T2DM (84.2% vs. 81.8%, *p* < 0.05), use of insulin (70.8% vs. 73.3%, *p* < 0.05), and use of SGLT2 inhibitors (28.6% vs. 23.5%, *p* < 0.001, Table 1).

### 3.2. Follow-Up Time

Among all patients, median follow-up was longer in the non-GLP-1 RA group than in the GLP-1 RA cohort both before and after PSM. Prior to matching, median follow-up was 1227 days among non-users (IQR, 1343) and 700.5 days in the GLP-1 RA users group (IQR, 907) (Figure 2, Table 3). After PSM, median follow up was 1101 days in the non-GLP-1 RA users group (IQR, 1343) and 702.5 days in the GLP-1 RA users group (IQR, 909) (Figure 2, Table 3).

### 3.3. Pancreatic Cancer Incidence in Unmatched and Matched Cohorts

Prior to matching, the incidence of pancreatic cancer was lower in the group that received a GLP-1 RA (hazard ratio (HR) 0.62, 95% confidence interval (CI) 0.41–0.95, *p* < 0.05, Figure 2, Table 3). This remained true after matching with lower pancreatic cancer incidence in the group that had received a GLP-1 RA (HR 0.49, 95% CI 0.30–0.80, *p* < 0.005, Figure 2, Table 3).

In a secondary analysis evaluating pancreatic occurrence, GLP-1 RA use was associated with lower odds of pancreatic cancer both before and after PSM. Prior to matching, pancreatic cancer occurred in 22 (0.7%) GLP-1 RA users and 1284 (1.5%) non-users (odds ratio (OR) 0.46, 95% CI 0.30–0.70, *p* < 0.001) with an absolute risk difference of −0.8% (95% CI −1.1% to −0.51%) (Figure 2, Table 4). After matching, GLP-1 RA use remained associated with lower odds of pancreatic cancer (OR 0.36, 95% CI 0.22–0.58, *p* < 0.001), Figure 2, Table 4).

### 3.4. Characteristics of Patients with Coexisting T2DM

Among 37,543 patients with both CP and T2DM, there were 2534 patients who were prescribed a GLP-1 RA (Table 2, Figure 1). Compared to patients who did not receive a GLP-1 RA, those who did were on average older (58.1 vs. 57.3 years), more often female (47.2% vs. 41.2%), greater average BMI (32.7 vs. 27.4), higher proportion with BMI ≥ 30 (67.5% vs. 38.1%), greater average HbA1c (8.5 vs. 7.8), and had higher proportion with HbA1c ≥ 8 (68% vs. 31.9%, *p* < 0.001) (Table 2). GLP-1 users were more likely to have a history of pancreatic cysts (18.2% vs. 15.2%), hypertension (87.5% vs. 73%), obesity (60.9% vs. 26.1%), or tobacco use (16.7% vs. 11.4%, *p* < 0.001, Table 2). Patients who received a GLP-1 RA also were more likely to receive insulin (81% vs. 62.3%), metformin (71.1% vs. 31.3%), or an SGLT2 inhibitor (34.1% vs. 5.6%, *p* < 0.001, Table 2).

After PSM, 2521 patients were included in the GLP-1 RA recipient group and 2521 were included in the group who did not receive a GLP-1 RA (Figure 1). After matching, all variables were balanced, except for average BMI (32.7 vs. 30.4, *p* < 0.001) and hyperlipidemia (74.5% vs. 77.6%, *p* < 0.05) (Table 2).

### 3.5. Follow-Up Time in the Patients with Coexisting T2DM

Among all patients with CP and T2DM, median follow-up was longer in the non-GLP-1 RA group than in the GLP-1 RA cohort both before and after PSM. Median follow up time was 1116 days among non-users (IQR, 1382) and 774 days among GLP-1 RA users (IQR, 962) (Figure 2, Table 3). After PSM, median follow-up was 1028 days in non-users (IQR, 1413) and 775 days in users (IQR, 966) (Figure 2, Table 3).

### 3.6. Pancreatic Cancer Incidence in Patients with CP and T2DM

Prior to matching, the 5-year incidence of pancreatic cancer among patients with CP and T2DM was lower in the group that received a GLP-1 RA (HR 0.60, 95% CI 0.39–0.92, *p* < 0.05, Figure 2, Table 3). This remained true after matching with lower pancreatic cancer incidence in the group that was in receipt of GLP-1 RA (HR 0.53, 95% CI 0.31–0.91, *p* < 0.05, Figure 2, Table 3).

In a secondary analysis evaluating pancreatic occurrence among patients with CP and T2DM, GLP-1 RA use was associated with lower odds of pancreatic cancer both before and after PSM. Prior to matching, pancreatic cancer occurred in 21 (0.8%) GLP-1 RA users and 578 (1.7%) non-users (OR 0.31–0.73, 95% CI 0.30–0.70, *p* < 0.001) with an absolute risk difference of −0.87% (95% CI −1.23% to −0.50%) (Figure 2, Table 4). After matching, GLP-1 RA use remained associated with lower odds of pancreatic cancer (OR 0.44, 95% CI 0.26–0.75, *p* < 0.01) (Figure 2, Table 4).

## 4. Discussion

In this analysis, we demonstrate lower 5-year incidence of pancreatic cancer development among a high-risk population of patients with CP who received a GLP-1 RA compared to those who did not. To our knowledge, this is one of the largest studies to assess incidence of pancreatic cancer by receipt of GLP-1 RAs in this population. Further, we show that this reduction in incidence is also true in patients with both CP and T2DM.

Multiple mechanisms may explain why we observed lower incidence of pancreatic cancer among patients receiving GLP-1 RAs. Through activation of the GLP-1 receptor, GLP1-RAs trigger intracellular signals that block oncogenic pathways supporting cancer cell growth and survival (ERK/MAPK, PI3K/Akt, NF- κB), activate AMPK while suppressing mTOR, and promote apoptosis and cell cycle arrest [18,19]. GLP-1 RAs have also been shown to impair tumor invasion and metastasis by regulating cell movement, adhesion, and tissue modeling [1]. In addition, the endogenous GLP-1 hormone has been shown to have anti-inflammatory, antioxidative, and vascular protective effects in various cells and tissues including the pancreas [2,3,4]. Many studies have reported preventive effects of GLP-1 and GLP-1 RAs on beta cells against various toxic stimuli, including cytokines and reactive oxygen species [18,19]. These antitumor mechanisms are especially important in the context of CP where persistent inflammation and oxidative stress contribute to DNA damage, genomic instability, and a pro-tumorigenic environment. Together, these mechanisms can explain why there may be a reduced risk of pancreatic cancer in patients with chronic pancreatitis who use GLP-1 RAs.

T2DM itself is an independent risk factor for cancer, with studies demonstrating a 73% higher risk of cancer compared with the general population [20]. There is also a link between T2DM and pancreatic cancer, with studies showing a 6–8-fold-higher risk of developing pancreatic cancer and a 2-fold increased risk for pancreatic-cancer-related mortality [15,16]. In T2DM, insulin resistance and hyperinsulinemia are responsible for stimulating the IGF-1 pathway, which in turn leads to the activation of the PI3K/mTOR pathways, thus favoring pancreatic cancer initiation, malignant transformation, and metastasis [21]. Further, inflammation in T2DM contributes to pancreatic carcinogenesis by promoting oxidative stress, activating the oncogenic pathways of NF-κB and STAT3, and can drive tumor promoting processes such as apoptosis inhibition, cell cycle progression, and epithelial–mesenchymal transition [21].

T2DM is a common comorbidity of CP with a point prevalence of approximately 30–40% [17]. Given the concurrent and overlapping risk of pancreatic cancer in patients with CP and T2DM, these patients represent a particularly vulnerable population for the development of pancreatic cancer. Historically, there have been concerns regarding pancreatic safety and the use of GLP-1 RA’s [22,23,24]. However, large-scale studies have provided reassurance. Ayoub et al. (2025) did a retrospective analysis with 61 healthcare organizations in the United States and found that in patients with T2DM, the lifetime risk of developing pancreatitis was lower in GLP-1 RA users (0.3% vs. 0.4%, *p* < 0.001) [25]. Cao et al. (2020) conducted a meta-analysis with over 56,000 patients with T2DM and found no significant difference in pancreatic cancer incidence compared to placebo with the use of GLP-1 RA’s (OR 1.12, 95% CI: 0.77–1.63, *p* = 0.56) [26]. Similarly, Wen et al. (2025) conducted a comprehensive meta-analysis of 62 randomized controlled trials involving over 66,000 patients with T2DM receiving GLP-1 RA based medications and their analysis found no significant association between medication usage and the risk of pancreatic cancer (RR 1.30, 95% CI 0.86–1.97, *p* = 0.22) [27]. These findings establish a crucial baseline that GLP-1 RA’s are safe or at least neutral in the large, high-risk T2DM population. This provides validity to our study that examines the risk of pancreatic cancer in an especially higher risk cohort of patients with CP and T2DM.

In our retrospective propensity score-matched multicenter study, we observed that GLP-1 RA use was associated with a significantly lower incidence of pancreatic cancer in patients with CP and concomitant T2DM. In a retrospective analysis performed by Henney et al. (2025), they also found that treatment with GLP-1 RA reduced the risk of pancreatic cancer in patients with T2DM (HR 0.73, 95% CI 0.63–0.85, *p* < 0.001) [28]. With inflammation being tightly regulated by NF-κB and STAT3 pathways, GLP-1 RA’s have been shown to inhibit NF-κB and reduce the generation of inflammatory mediators, among other inflammatory cytokines, reactive oxygen species, adhesion molecules, and chemokines [18,19,29]. In addition, GLP-1 RA’s exert their effects on regulating insulin release, suppressing glucagon secretion, and slowing gastric emptying [30]. In doing this, these medications can reduce the activation of the IGF-1 pathways as well as reduce the inflammation that contributes to oxidative stress and promotes pancreatic cancer in patients with T2DM.

The aforementioned studies primarily established drug neutrality and safety regarding the pancreas in patients with T2DM. Our study demonstrates a stronger and unique effect of GLP-1 RAs in patients with CP and concomitant T2DM. The therapeutic mechanisms of GLP-1 RA based medications are not only optimally suited to neutralize the pro-carcinogenic inflammation and oxidative stress in CP, but also limits the metabolic risk in T2DM. This results in a statistically significant reduction in pancreatic cancer in these high-risk populations with the use of GLP-1 RA’s.

Limitations of our study include the use of billing codes and retrospective cohort analysis. Additionally, differences in index date definitions across cohorts may result in longer observation time for the control group, potentially increasing their opportunity to develop pancreatic cancer and introducing immortal time bias. Further, adherence to GLP-1 RA therapy, duration of therapy, and dosage of medications were unable to be assessed. In addition, although PSM was performed for available characteristics, several confounding variables exist that cannot be accounted for based on available data, including lifestyle factors, such as diet and physical activity, and other clinical characteristics, such as severity of chronic pancreatitis and genetic predisposition/family history of pancreatic cancer. Additionally, in our analysis, the GLP-1 RA group had a higher concomitant use of SGLT-2 inhibitors and given the emerging evidence that these medications may have some anti-neoplastic effect, this is a potential confounding factor. We were also unable to account for whether patients switched between different GLP-1 RA’s during the study period, which may have introduced heterogeneity in exposure. Further, GLP-1 RAs were assessed as a class, which may obscure drug specific impact on cancer risk. Lastly, a relatively short follow-up period and low incidence of pancreatic cancer require cautious interpretation of these findings and replication of this study in the future with longer available follow-up will be important.

## 5. Conclusions

The intersection of CP-related inflammatory injury and T2DM-mediated metabolic stress creates a high-risk environment that promotes the development of pancreatic cancer, and our findings suggest an association between GLP-1 RA therapy and a reduction in risk of pancreatic cancer in this population. Further studies should aim to evaluate these findings in a prospective setting.

## Figures and Tables

**Figure 1 cancers-18-00179-f001:**
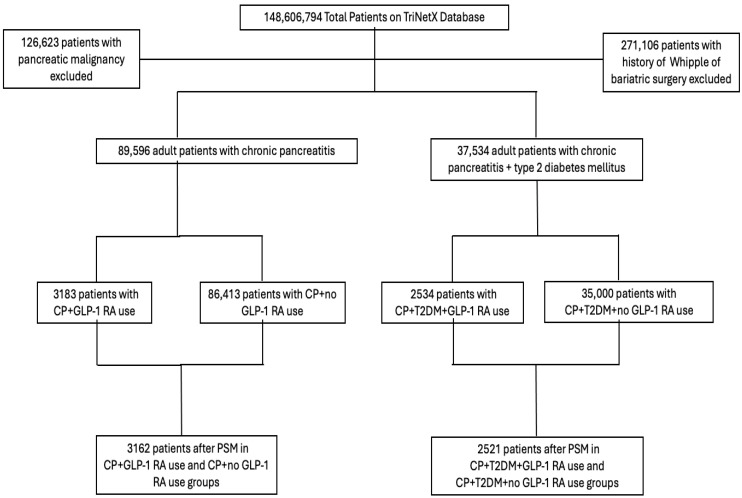
Patients were stratified by diagnosis of chronic pancreatitis (CP) alone or CP with type 2 diabetes mellitus (T2DM) and further stratified by glucagon-like peptide-1 receptor agonist (GLP-1 RA) use; propensity score matching (PSM) was used to generate the final analytic cohort.

**Figure 2 cancers-18-00179-f002:**
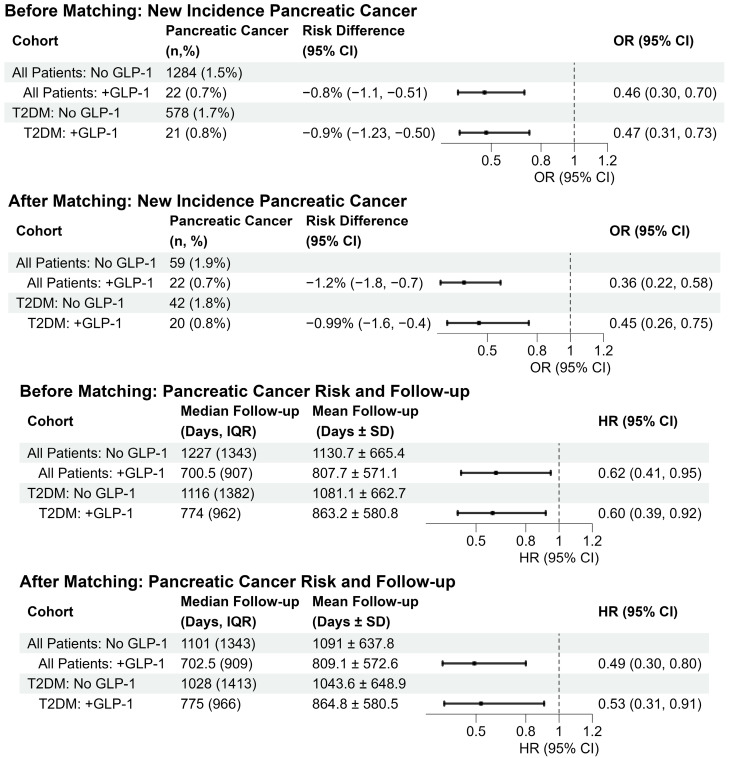
Forest plots showing the odds ratios (OR) for pancreatic cancer incidence and hazard ratios (HR) for pancreatic cancer risk in patients with chronic pancreatitis, stratified by all patients and patients with type 2 diabetes mellitus (T2DM), before and after propensity score matching. Symbols represent the estimated odds ratio (OR) or hazard ratio (HR), and horizontal lines indicate the 95% confidence intervals (CIs). The “No GLP-1” cohorts are the reference for all comparisons. Estimates with 95% CIs that do not cross the reference line at 1.0 are considered statistically significant (*p* < 0.05).

**Table 1 cancers-18-00179-t001:** Baseline characteristics of adult chronic pancreatitis (CP) patients stratified by glucagon-like peptide-1 receptor agonist (GLP-1 RA) usage, before and after propensity score matching.

	Before Matching			After Matching		
Characteristic Name	No GLP-1 ^1^	+GLP-1 ^1^	SMD ^2^	No GLP-1 ^1^	+GLP-1 ^1^	SMD ^2^
n	86,413	3183		3162	3162	
**Age at Index**	54.58 ± 15.38	57.28 ± 12.79	0.19 ***	57.7 ± 13.85	57.27 ± 12.79	0.03
**Sex**						
Male	46,261 (53.5%)	1531 (48.1%)	0.11 ***	1469 (46.5%)	1527 (48.3%)	0.04
Female	40,133 (46.4%)	1652 (51.9%)	0.11 ***	1693 (53.5%)	1635 (51.7%)	0.04
**Race**						
American Indian or Alaska Native	574 (0.7%)	29 (0.9%)	0.03	25 (0.8%)	29 (0.9%)	0.01
Asian	2609 (3%)	103 (3.2%)	0.01	107 (3.4%)	103 (3.3%)	0.01
Black or African American	14,618 (16.9%)	550 (17.3%)	0.01	557 (17.6%)	547 (17.3%)	0.01
Native Hawaiian or Other Pacific Islander	327 (0.4%)	18 (0.6%)	0.03	18 (0.6%)	18 (0.6%)	0.00
Other Race	2550 (3%)	144 (4.5%)	0.08 ***	122 (3.9%)	141 (4.5%)	0.03
Unknown Race	6519 (7.5%)	160 (5%)	0.10 ***	135 (4.3%)	159 (5%)	0.04
White	59,216 (68.5%)	2179 (68.5%)	0.00	2198 (69.5%)	2165 (68.5%)	0.02
**Ethnicity**						
Hispanic or Latino	5150 (6%)	285 (9%)	0.11 ***	274 (8.7%)	282 (8.9%)	0.01
Not Hispanic or Latino	63,795 (73.8%)	2440 (76.7%)	0.07 ***	2443 (77.3%)	2423 (76.6%)	0.02
Unknown Ethnicity	17,468 (20.2%)	458 (14.4%)	0.15 ***	445 (14.1%)	457 (14.5%)	0.01
**Labs**						
BMI (mean) ^3^	26.36 ± 6.49	33.26 ± 7.73	0.97 ***	31.11 ± 7.12	33.26 ± 7.73	0.29 ***
BMI ≥ 30	24,102 (27.9%)	2215 (69.6%)	0.92 ***	2277 (72%)	2194 (69.4%)	0.06 *
Hemoglobin A1c (mean) ^4^	6.8 ± 2.14	8.09 ± 2.23	0.59 ***	8.08 ± 2.3	8.08 ± 2.23	0.00
Hemoglobin A1c ≥ 8	10,059 (11.6%)	1782 (56%)	1.06 ***	1771 (56%)	1761 (55.7%)	0.01
**Comorbidities**						
Alcohol-related disorders	21,490 (24.9%)	681 (21.4%)	0.08 ***	632 (20%)	675 (21.3%)	0.03
Cyst of pancreas	10,451 (12.1%)	590 (18.5%)	0.18 ***	564 (17.8%)	580 (18.3%)	0.01
Hypertension	45,345 (52.5%)	2642 (83%)	0.69 ***	2686 (84.9%)	2622 (82.9%)	0.06 *
Hyperlipidemia	27,192 (31.5%)	2202 (69.2%)	0.81 ***	2248 (71.1%)	2185 (69.1%)	0.04
Overweight/obesity	14,044 (16.3%)	2010 (63.1%)	1.09 ***	2104 (66.5%)	1989 (62.9%)	0.08 **
Tobacco use	7950 (9.2%)	499 (15.7%)	0.20 ***	474 (15%)	491 (15.5%)	0.01
Type 2 Diabetes	27,683 (32%)	2682 (84.3%)	1.25 ***	2586 (81.8%)	2661 (84.2%)	0.06 *
**Medications**						
Insulin	23,568 (27.3%)	2260 (71%)	0.97 ***	2319 (73.3%)	2239 (70.8%)	0.06 *
Metformin	10,154 (11.8%)	1983 (62.3%)	1.23 ***	1944 (61.5%)	1962 (62%)	0.01
SGLT2 inhibitors	1945 (2.3%)	924 (29%)	0.79 ***	743 (23.5%)	903 (28.6%)	0.12 ***
**GLP-1 RA Medications**						
Albiglutide	0 (0%)	10 (0.3%)	NA	0 (0%)	10 (0.3%)	NA
Dulaglutide	0 (0%)	844 (26.5%)	NA	0 (0%)	836 (26.4%)	NA
Exenatide	0 (0%)	58 (1.8%)	NA	0 (0%)	58 (1.8%)	NA
Liraglutide	0 (0%)	332 (10.4%)	NA	0 (0%)	331 (10.5%)	NA
Lixisenatide	0 (0%)	38 (1.2%)	NA	0 (0%)	38 (1.2%)	NA
Semaglutide	0 (0%)	1469 (46.2%)	NA	0 (0%)	1461 (46.2%)	NA
Tirzepatide	0 (0%)	459 (14.4%)	NA	0 (0%)	455 (14.4%)	NA

^1^ Mean ± SD; n (%). ^2^ SMD: Standardized mean difference; SMD < 0.1 considered balanced. *p*-values from independent *t*-tests (continuous variables) or chi-squared tests (categorical variables) are indicated as follows: * *p* < 0.05, ** *p* < 0.01, *** *p* < 0.001. ^3^ BMI: +GLP-1 n = 2721 n = 2277 (after matching); No GLP-1 n = 26,706; n = 2529 (after matching). ^4^ Hemoglobin A1c: +GLP-1 n = 2769; n = 2748 (after matching); No GLP-1 n = 61,578; n = 2763 (after matching); NA: not applicable.

**Table 2 cancers-18-00179-t002:** Baseline characteristics of adult chronic pancreatitis (CP) patients with Type 2 Diabetes (T2DM) stratified by glucagon-like peptide-1 receptor agonist (GLP-1 RA) usage, before and after propensity score matching.

	Before Matching			After Matching		
	No GLP-1 ^1^	+GLP-1 ^1^	SMD ^2^	No GLP-1 ^1^	+GLP-1 ^1^	SMD ^2^
n	35,000	2534		2521	2521	
**Age at Index**	57.25 ± 14.35	58.09 ± 12.57	0.06 **	58.59 ± 13.49	58.09 ± 12.56	0.04
**Sex**						
Male	20,559 (58.7%)	1339 (52.8%)	0.12 ***	1314 (52.1%)	1335 (53%)	0.02
Female	14,435 (41.2%)	1195 (47.2%)	0.12 ***	1207 (47.9%)	1186 (47%)	0.02
**Race**						
American Indian or Alaska Native	289 (0.8%)	24 (0.9%)	0.01	17 (0.7%)	24 (1%)	0.03
Asian	1240 (3.5%)	93 (3.7%)	0.01	95 (3.8%)	93 (3.7%)	0.00
Black or African American	7292 (20.8%)	491 (19.4%)	0.04	504 (20%)	490 (19.4%)	0.01
Native Hawaiian or Other Pacific Islander	173 (0.5%)	17 (0.7%)	0.02	16 (0.6%)	17 (0.7%)	0.00
Other Race	1031 (2.9%)	115 (4.5%)	0.08 ***	100 (4%)	109 (4.3%)	0.02
Unknown Race	2454 (7%)	143 (5.6%)	0.06 **	129 (5.1%)	143 (5.7%)	0.02
White	22,521 (64.3%)	1651 (65.2%)	0.02	1660 (65.8%)	1645 (65.3%)	0.01
**Ethnicity**						
Hispanic or Latino	2504 (7.2%)	247 (9.7%)	0.09 ***	247 (9.8%)	243 (9.6%)	0.01
Not Hispanic or Latino	26,102 (74.6%)	1908 (75.3%)	0.02	1905 (75.6%)	1900 (75.4%)	0.00
Unknown Ethnicity	6394 (18.3%)	379 (15%)	0.09 ***	369 (14.6%)	378 (15%)	0.01
**Labs**						
BMI ^3^	27.36 ± 6.98	32.73 ± 7.74	0.73 ***	30.4 ± 7.18	32.71 ± 7.73	0.31 ***
BMI ≥ 30	13,318 (38.1%)	1711 (67.5%)	0.62 ***	1700 (67.4%)	1698 (67.4%)	0.00
Hemoglobin A1c ^4^	7.75 ± 2.33	8.5 ± 2.13	0.33 ***	8.37 ± 2.25	8.5 ± 2.14	0.06
Hemoglobin A1c ≥ 8	11,158 (31.9%)	1722 (68%)	0.77 ***	1731 (68.7%)	1709 (67.8%)	0.02
**Comorbidities**						
Alcohol-related disorders	9177 (26.2%)	561 (22.1%)	0.10 ***	539 (21.4%)	558 (22.1%)	0.02
Cyst of pancreas	5333 (15.2%)	460 (18.2%)	0.08 ***	444 (17.6%)	451 (17.9%)	0.01
Hypertension	25,544 (73%)	2218 (87.5%)	0.37 ***	2233 (88.6%)	2205 (87.5%)	0.03
Hyperlipidemia	17,381 (49.7%)	1892 (74.7%)	0.53 ***	1956 (77.6%)	1879 (74.5%)	0.07 *
Overweight/obesity (E66)	9136 (26.1%)	1543 (60.9%)	0.75 ***	1531 (60.7%)	1530 (60.7%)	0.00
Tobacco use	3986 (11.4%)	424 (16.7%)	0.15 ***	405 (16.1%)	421 (16.7%)	0.02
**Medications**						
Insulin	21,798 (62.3%)	2053 (81%)	0.43 ***	2074 (82.3%)	2042 (81%)	0.03
Metformin	10,968 (31.3%)	1802 (71.1%)	0.87 ***	1830 (72.6%)	1789 (71%)	0.04
SGLT2 inhibitors	1969 (5.6%)	865 (34.1%)	0.76 ***	788 (31.3%)	852 (33.8%)	0.05
**GLP-1 RA Medications**						
Albiglutide	0 (0%)	10 (0.4%)	NA	0 (0%)	10 (0.4%)	NA
Dulaglutide	0 (0%)	786 (31%)	NA	0 (0%)	782 (31%)	NA
Exenatide	0 (0%)	50 (2%)	NA	0 (0%)	50 (2%)	NA
Liraglutide	0 (0%)	278 (11%)	NA	0 (0%)	277 (11%)	NA
Lixisenatide	0 (0%)	33 (1.3%)	NA	0 (0%)	33 (1.3%)	NA
Semaglutide	0 (0%)	1119 (44.2%)	NA	0 (0%)	1113 (44.1%)	NA
Tirzepatide	0 (0%)	283 (11.2%)	NA	0 (0%)	281 (11.1%)	NA

^1^ Mean ± SD; n (%). ^2^ SMD: Standardized mean difference; SMD < 0.1 considered balanced. *p*-values from independent *t*-tests (continuous variables) or chi-squared tests (categorical variables) are indicated as follows: * *p* < 0.05, ** *p* < 0.01, *** *p* < 0.001. ^3^ BMI: + GLP-1 n = 2291; n = 2225 (after matching); No GLP-1 n = 26,706; n = 2175 (after matching). ^4^ Hemoglobin A1c: +GLP1 n = 2154; n = 2175 (after matching); No GLP-1 n = 22,810; n = 2225 (after matching); NA: Not applicable.

**Table 3 cancers-18-00179-t003:** Five-year incidence of pancreatic cancer in patients with chronic pancreatitis (CP) and in patients with CP plus type 2 diabetes mellitus (T2DM), stratified by GLP-1 RA exposure, before and after propensity score matching. Hazard ratios (HRs) with 95% confidence intervals and median follow-up times for each cohort are reported.

	Before Matching				After Matching			
	Mean Follow-Up (Days ± SD)	Median Follow-Up Days (IQR)	HR (95% CI)	*p*-Value	Mean Follow-Up (Days + SD)	Median Follow-Up Days (IQR)	HR (95% CI)	*p*-Value
**All Patients**								
No GLP-1 vs. +GLP-1	1130.7 ± 665.4	1227 (1343)			1091 ± 637.8	1101 (1343)		
	807.7 ± 571.1	700.5 (907)	0.62 (0.41, 0.95)	0.03	809.1 ± 572.6	702.5 (909)	0.49 (0.30, 0.80)	0.003
**Type 2 Diabetic Patients**								
No GLP-1 vs. + GLP-1	1081.1 ± 662.7	1116 (1382)			1043.6 ± 648.9	1028 (1413)		
	863.2 ± 580.8	774 (962)	0.60 (0.39, 0.92)	0.02	864.8 ± 580.5	775 (966)	0.53 (0.31, 0.91)	0.02

**Table 4 cancers-18-00179-t004:** 5-year incidence of pancreatic cancer in patients with chronic pancreatitis and patients with chronic pancreatitis and concomitant type 2 diabetes mellitus, with and without GLP-1RA exposure, before and after propensity score matching.

	Before Matching				After Matching			
	N (%)	Risk Difference (95% CI)	Odds Ratio (95% CI)	*p*-Value	*n* (%)	Risk Difference (95% CI)	Odds Ratio (95% CI)	*p*-Value
All Patients								
No GLP-1 vs. +GLP-1	1284 (1.5%)				59 (1.9)			
	22 (0.7%)	−0.8% (−1.1, −0.51)	0.46 (0.30, 0.70)	<0.001	22 (0.7)	−1.24% (−1.81, −0.67)	0.36 (0.22, 0.58)	<0.001
**Type 2 Diabetic Patients**								
No GLP-1 vs. +GLP-1	578 (1.7%)				42 (1.8%)			
	21 (0.8%)	−0.87% (−1.23, −0.50)	0.47 (0.31, 0.73)	<0.001	20 (0.8%)	−0.99% (−1.6, −0.36)	0.44 (0.26, 0.75)	0.002

## Data Availability

The data used for this manuscript is publicly available. Further inquiries can be directed to the corresponding author.

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
