# Peer review of "Glucagon-like Peptide-1 Receptor Agonist Use and Pancreatic Cancer Risk in Patients with Chronic Pancreatitis"

_cancers, 2026, doi:10.3390/cancers18020179_

Round 1

Reviewer 1 Report

Comments and Suggestions for Authors

Although the study provides valuable observational evidence that is hypothesis-generating for future research, the study has several important limitations that should be acknowledged.

First, as a retrospective cohort analysis, residual confounding and selection bias cannot be fully excluded despite the use of propensity score matching. Unmeasured factors—such as the severity of chronic pancreatitis, genetic predisposition, and family history of pancreatic cancer, dietary patterns, and other lifestyle variables—may have influenced the observed associations.

Second, all GLP-1 receptor agonists were analyzed as a single class, which may obscure drug-specific effects.

Third, the relatively short 5-year follow-up period .

Finally, the low absolute incidence of pancreatic cancer (<2%) limits statistical power and warrants cautious interpretation of the findings.

Author Response

Comment 1: Although the study provides valuable observational evidence that is hypothesis-generating for future research, the study has several important limitations that should be acknowledged.

First, as a retrospective cohort analysis, residual confounding and selection bias cannot be fully excluded despite the use of propensity score matching. Unmeasured factors—such as the severity of chronic pancreatitis, genetic predisposition, and family history of pancreatic cancer, dietary patterns, and other lifestyle variables—may have influenced the observed associations.

Response 1: We agree with the insightful comments. These factors are certainly limitations of the current analysis, and we have now included these in the limitations within the discussion section.

Comment 2: Second, all GLP-1 receptor agonists were analyzed as a single class, which may obscure drug-specific effects.

Response 2: We analyzed GLP-1 RAs as a single class, in order to increase patient number to add statistical power. We make mention of this as a limitation in the discussion, but we hope we are able to demonstrate the possibility of reduction in pancreatic cancer incidence associated with GLP-1 RAs as a class. At this time, we do not have reason to believe there would be a unique property among one drug versus another that would afford a more protective benefit in terms of pancreatic cancer risk.

Comment 3: Third, the relatively short 5-year follow-up period.

Response 3: This is now noted in the limitations section of the manuscript, noting the importance of replication of this study in the future when longer follow-up data is available.

Comment 4: Finally, the low absolute incidence of pancreatic cancer (<2%) limits statistical power and warrants cautious interpretation of the findings.

Response 4: We have now included this limitation in the discussion section now, and coincides with our statement regarding the importance of longer follow-up, when this data becomes available.

Reviewer 2 Report

Comments and Suggestions for Authors

Authors have evaluated the association between GLP-1 RA use and pancreatic cancer incidence among patients with chronic pancreatitis (CP), and among those with CP and T2DM  using  TriNetX database in the United States. GLP-1 RA use was associated with a lower 5-year incidence of pancreatic cancer (OR) 0.39.

This is a clinically relevant study. However, there are some serious methodological issues:

  • You investigated incidence of pancreatic cancer but you did not use K-M-curves and not Cox regression, but logistic regression with Ors. This is methodologically suboptimal. The follow-up can be different among cohorts what cause more cancer cases in non-G;LP1 cohort.   Authors explained that database does not natively provide Cox proportional hazards modelling- but that does not change the fact that with this method no plausible results can be achieved.
  • As absolute event rate is very low, each change in event number (+ - 2 ev ents) can cause fully different findings. Especially, using a wrong method, this may be fatal for interpretation.
  • Pancreatitis severity is lacking- minor issue but very relevant when cancer risk is evaluated.

Author Response

Authors have evaluated the association between GLP-1 RA use and pancreatic cancer incidence among patients with chronic pancreatitis (CP), and among those with CP and T2DM  using  TriNetX database in the United States. GLP-1 RA use was associated with a lower 5-year incidence of pancreatic cancer (OR) 0.39.

This is a clinically relevant study. However, there are some serious methodological issues:

Comment 1: You investigated incidence of pancreatic cancer but you did not use K-M-curves and not Cox regression, but logistic regression with Ors. This is methodologically suboptimal. The follow-up can be different among cohorts what cause more cancer cases in non-G;LP1 cohort.   Authors explained that database does not natively provide Cox proportional hazards modelling- but that does not change the fact that with this method no plausible results can be achieved.

Response 1: Thank you for your suggestion. We have now incorporated Kaplan–Meier analyses to evaluate time to pancreatic cancer, reporting hazard ratios with 95% confidence intervals within a predefined time window of 6 months to 5 years after the index date. These analyses also provide cohort-specific follow-up information, which has been incorporated into the Results section. We apologize for the confusion regarding the modeling capabilities of the TriNetX database, as Cox proportional hazard modeling is available as an analytical tool.

Comment 2: As absolute event rate is very low, each change in event number (+ - 2 ev ents) can cause fully different findings. Especially, using a wrong method, this may be fatal for interpretation.

Response 2: We agree with the reviewer that analyses of rare outcomes such as pancreatic cancer are inherently sensitive to small absolute changes in event counts, and that even a difference of only a few events can meaningfully affect model estimates. This is a recognized challenge in epidemiological studies of low-incidence cancers and applies regardless of the statistical method used. Importantly, the low absolute event rate observed in our study is not a consequence of using logistic regression but rather reflects the biological rarity of pancreatic cancer in the studied population and the length of follow up. Any modeling approach applied to sparse data, including logistic, Cox, or Poisson models, will exhibit instability with small changes in event numbers. Time-to-event methods do not necessarily eliminate this limitation; they instead express the association on a different scale.

We also emphasize that strict inclusion criteria were applied, particularly with respect to follow-up and diagnostic accuracy, to ensure appropriate representation of the study population. As a result, the observed incidence is likely conservative and may underestimate the true association between GLP-1 exposure and pancreatic cancer risk.

Comment 3: Pancreatitis severity is lacking- minor issue but very relevant when cancer risk is evaluated.

Response 3: Reviewer 1 has also noted this limitation, and we now mention this as limitation in the discussion section of the manuscript.

Reviewer 3 Report

Comments and Suggestions for Authors

Thank you for the opportunity to review this article. This original study evaluates the association between GLP-1 RA use and pancreatic cancer incidence among patients with chronic pancreatitis and those with both chronic pancreatitis and diabetes mellitus. The study is well designed, and the methods are presented appropriately. The results are interesting; however, I agree that further prospective studies are necessary. I have only minor issues to address before reconsideration.

  1. The manuscript is generally well written, although it contains some minor grammatical and syntactical errors.
  2. The Introduction would benefit from including some epidemiological information.
  3. In the Figure 1 caption, all abbreviations used in the diagram should be spelled out.
  4. Similarly, all abbreviations used in the tables should be expanded in the captions to improve readability.
  5. The discussion should be expanded, and authors should attempt to find similar articles in the field to compare them.
  6. The study limitations should be elaborated upon, including additional confounders such as family history of cancer, among others.

Author Response

Comment 1: The manuscript is generally well written, although it contains some minor grammatical and syntactical errors.

Response 1: Grammatical and syntactical errors are now fixed.

Comment 2: The Introduction would benefit from including some epidemiological information.

Response 2: Thanks for the suggestion. Epidemiologic information is now mentioned in the introduction.

Comment 3: In the Figure 1 caption, all abbreviations used in the diagram should be spelled out.

Response 3: All abbreviations are now listed in the caption of Figure 1.

Comment 4: Similarly, all abbreviations used in the tables should be expanded in the captions to improve readability.

Response 4: All abbreviations used in the tables are now expanded in the captions.

Comment 5: The discussion should be expanded, and authors should attempt to find similar articles in the field to compare them

Response 5: As suggested, other articles are now referenced in the discussion.

Comment 6: The study limitations should be elaborated upon, including additional confounders such as family history of cancer, among others.

Response 6: Reviewers 1 and 2 also noted these limitations and are now mentioned in the discussion section of the manuscript.

Reviewer 4 Report

Comments and Suggestions for Authors

This manuscript addresses an important and timely clinical question with a large multi-center dataset and suggests a potentially meaningful association between GLP-1 RA therapy and reduced pancreatic cancer incidence in patients with chronic pancreatitis, including those with T2DM. However, several methodological issues (particularly around index date definition, potential immortal time bias, handling of follow-up time, and residual confounding) need to be clarified and, where possible, addressed analytically.

  1. For the GLP-1 cohort, the index date is defined as GLP-1 RA initiation; for the non-GLP-1 cohort, it is the first recorded instance of CP.  This design creates a period between CP diagnosis and GLP-1 initiation during which GLP-1-treated patients must remain alive and pancreatic-cancer–free in order to receive the exposure, i.e., “immortal time.” Please clarify the median time from CP diagnosis to GLP-1 RA initiation in exposed patients, how long after CP diagnosis non-exposed patients are followed, and whether any “lag” or wash-in period was applied symmetrically. This potential immortal time bias must be explicitly discussed as a major limitation, and a sensitivity analysis that (for example) re-indexes non-users at a comparable time after CP diagnosis would be helpful.

  1. The primary outcome is “5-year incidence of pancreatic cancer from 6 months up to 5 years after the index date,” but the analysis appears to use logistic regression/odds ratios without accounting for variation in follow-up duration or censoring. Please provide median (IQR) follow-up time for each cohort and subgroup, and B) Kaplan–Meier or cumulative incidence curves for pancreatic cancer by exposure group. A Cox proportional hazards model (or a Fine–Gray model if competing risk of death is substantial) on the matched cohorts would be more appropriate than simple odds ratios and would allow explicit handling of censoring. If this is not possible within TriNetX, please state this clearly and justify the chosen approach.

  1. After PSM, several covariates remain imbalanced between GLP-1 and non-GLP-1 groups (e.g., BMI, HbA1c, SGLT2 inhibitor use in both the overall CP and CP+T2DM cohorts). Please report standardized mean differences (SMDs) for all matched variables, not just p-values, to better quantify post-matching balance. Also, consider an additional multivariable regression (e.g., conditional logistic regression or Cox model) within the matched sample, further adjusting for the remaining imbalanced covariates (BMI, HbA1c, and SGLT2 inhibitor use). Given emerging data on SGLT2 inhibitors and cancer outcomes, the higher use of SGLT2 inhibitors in the GLP-1 group (even after matching) should be explicitly discussed as a potential confounder.

  1. Footnotes in Table 1 and Table 3 indicate that BMI and HbA1c were missing for substantial proportions of patients (e.g., BMI available in 2,692/3,288 GLP-1 users vs 61,515/88,280 non-users before matching). Please clarify, how missing BMI and HbA1c values were handled in propensity score modelling (complete-case analysis, imputation, inclusion of missingness indicators, etc.). Also, clarify ehether patients without BMI/HbA1c data were excluded before matching, and how this might affect generalizability and introduce selection bias. A short sensitivity analysis excluding BMI/HbA1c from the matching variables or using multiple imputation (if supported) would strengthen the robustness of the findings.

  1. Patients selected for GLP-1 RA therapy are likely to differ systematically from non-users in ways not fully captured by your covariates (e.g., care by endocrinologists, frequency of visits, socioeconomic status, lifestyle factors, detailed alcohol/tobacco exposure, CP severity, imaging intensity). While some of this is acknowledged in the Limitations, I would encourage you to expand the discussion of channeling bias, particularly how closer follow-up and more imaging in GLP-1 users might be expected to increase cancer detection rather than reduce it (which would, if anything, bias toward the null or against your observed direction). Also, consider an active comparator sensitivity analysis (e.g., GLP-1 RA vs another second-line diabetes agent such as SGLT2 inhibitor or DPP-4 inhibitor) if TriNetX allows, which would partially mitigate confounding by indication.

  1. The absolute number of pancreatic cancer cases in the GLP-1 groups is small (e.g., 20 events vs 50 in the matched CP cohort; 19 vs 37 in the matched CP+T2DM cohort). Please report exact event counts clearly in the text (they are visible in Table 2, but emphasizing this in the Results would help contextualize the ORs) and provide absolute risk differences and, if feasible, numbers needed to treat (NNT) with appropriate confidence intervals. The Discussion and Conclusion should temper the language around a “protective effect,” emphasizing that this is an association observed in an observational study with relatively few events, rather than definitive evidence of causality.

  1. The Methods state that patients were excluded if they had prior GLP-1 RA exposure, prior pancreatic cancer, Whipple procedure, or bariatric surgery, and that at least two CP codes and one follow-up visit were required. Please clarify whether “at least two CP codes” had to occur prior to the index date and within a specified time window, and how recurrent or acute pancreatitis episodes were distinguished from chronic pancreatitis (K86.1 codes can be variably used). Also consider whether any minimum follow-up duration beyond the 6-month lag was enforced symmetrically in both groups. It would be helpful if Figure 1 could include the numbers and reasons for exclusions (prior pancreatic cancer, surgeries, missing key data) to increase transparency.

  1. You mention that large trials and observational studies have not consistently shown an association between GLP-1 RAs and pancreatic cancer. It would strengthen the paper to more systematically summarize the existing data specific to pancreatic cancer, distinguishing between safety signals (pancreatitis, pancreatic cancer) and emerging evidence of potential benefit. Please explicitly discuss how your findings in a high-risk CP/CP+T2DM population compare with findings in general T2DM populations (e.g., are your effect sizes larger, and if so, why might that be?).

  1. In Table 1 and Table 3, the section header “GLP-1 Inhibitors” is misleading; these are GLP-1 receptor agonists, not inhibitors. Please correct the terminology. In Table 2, the wording “No GLP-1 vs. +GLP1” is a bit confusing; consider separate rows “GLP-1 RA users” and “Non-users” with explicit labels.

  1. Standardize p-value reporting (e.g., use “p < 0.001” rather than “p=0.0002” for very small values, or follow the journal’s preferred convention). Ensure consistent decimal places for ORs and CIs across text and tables.

  1. Figure 1 (flowchart) is useful; consider slightly enlarging the font or simplifying the text boxes to improve readability. Indicating the numbers excluded at each step would improve transparency. You might consider adding a forest plot showing ORs (or HRs if time-to-event models are implemented) for CP overall and CP+T2DM, which would make the main results more visually accessible.

  1. You already note several limitations; after addressing the major points above, please ensure that immortal time bias / time-alignment issues are explicitly mentioned, potential misclassification of CP, T2DM, and pancreatic cancer diagnoses from administrative codes is acknowledged, and the absence of information on GLP-1 RA dose, duration, and adherence is clearly stated as a key limitation.

Author Response

Comment 1: For the GLP-1 cohort, the index date is defined as GLP-1 RA initiation; for the non-GLP-1 cohort, it is the first recorded instance of CP.  This design creates a period between CP diagnosis and GLP-1 initiation during which GLP-1-treated patients must remain alive and pancreatic-cancer–free in order to receive the exposure, i.e., “immortal time.” Please clarify the median time from CP diagnosis to GLP-1 RA initiation in exposed patients, how long after CP diagnosis non-exposed patients are followed, and whether any “lag” or wash-in period was applied symmetrically. This potential immortal time bias must be explicitly discussed as a major limitation, and a sensitivity analysis that (for example) re-indexes non-users at a comparable time after CP diagnosis would be helpful.

Response 1: We thank the reviewer for this insightful comment and agree that differing index date definitions introduce the potential for immortal time bias. This study represents an important initial analysis to explore potential associations between GLP-1 RA use and pancreatic cancer risk among patients with chronic pancreatitis, and we have explicitly noted the index date differences in the Methods and listed them as a study limitation.

Although we report follow-up time for each cohort, we were unable to quantify the median interval from chronic pancreatitis diagnosis to GLP-1 initiation due to limitations of the TriNetX platform. Accordingly, our findings should be interpreted as associative rather than causal. Future studies examining specific treatment pathways, including more granular therapy sequencing and time-varying exposure definitions, will be important to further clarify the relationship between GLP-1 use and pancreatic cancer risk in chronic pancreatitis patients.

Comment 2: The primary outcome is “5-year incidence of pancreatic cancer from 6 months up to 5 years after the index date,” but the analysis appears to use logistic regression/odds ratios without accounting for variation in follow-up duration or censoring. Please provide median (IQR) follow-up time for each cohort and subgroup, and B) Kaplan–Meier or cumulative incidence curves for pancreatic cancer by exposure group. A Cox proportional hazards model (or a Fine–Gray model if competing risk of death is substantial) on the matched cohorts would be more appropriate than simple odds ratios and would allow explicit handling of censoring. If this is not possible within TriNetX, please state this clearly and justify the chosen approach.

Response 2: We have now incorporated Kaplan–Meier analyses to evaluate time to pancreatic cancer, presented both before and after propensity score matching. In addition, we now report median (IQR) follow-up time for each cohort and subgroup, which has been incorporated into the Results section. Given this time-to-event framework, we also report hazard ratios with corresponding confidence intervals.

Comment 3: After PSM, several covariates remain imbalanced between GLP-1 and non-GLP-1 groups (e.g., BMI, HbA1c, SGLT2 inhibitor use in both the overall CP and CP+T2DM cohorts). Please report standardized mean differences (SMDs) for all matched variables, not just p-values, to better quantify post-matching balance. Also, consider an additional multivariable regression (e.g., conditional logistic regression or Cox model) within the matched sample, further adjusting for the remaining imbalanced covariates (BMI, HbA1c, and SGLT2 inhibitor use). Given emerging data on SGLT2 inhibitors and cancer outcomes, the higher use of SGLT2 inhibitors in the GLP-1 group (even after matching) should be explicitly discussed as a potential confounder.

Response 3: We agree that standardized mean differences (SMDs) provide a more appropriate and informative measure of quantifying covariate balance after propensity score matching than p-values. The description of SMDs was inadvertently omitted in the original submission.

We have now replaced p-values with SMDs for all matched variables in Tables 1 and 2 and added a brief description in the Methods section outlining the use of SMDs to assess post-matching balance.  Although some residual imbalance remains for select covariates, including BMI, HbA1c, and SGLT2 inhibitor use, overall covariate balance was substantially improved following matching.

We agree that a doubly robust approach, incorporating additional adjustment for remaining imbalanced covariates within a multivariable Cox proportional hazards model, would be methodologically ideal. However, this approach is not feasible within the constraints of the TriNetX online analytics platform.

SGLT-2’s as a potential confounding factor is now discussed in the limitations portion of the manuscript.

Comment 4: Footnotes in Table 1 and Table 3 indicate that BMI and HbA1c were missing for substantial proportions of patients (e.g., BMI available in 2,692/3,288 GLP-1 users vs 61,515/88,280 non-users before matching). Please clarify, how missing BMI and HbA1c values were handled in propensity score modelling (complete-case analysis, imputation, inclusion of missingness indicators, etc.). Also, clarify ehether patients without BMI/HbA1c data were excluded before matching, and how this might affect generalizability and introduce selection bias. A short sensitivity analysis excluding BMI/HbA1c from the matching variables or using multiple imputation (if supported) would strengthen the robustness of the findings.

Response 4: Within the TriNetX platform, laboratory variables included in propensity score matching cannot be modeled as continuous measures and are instead incorporated as user-defined categorical thresholds (e.g., BMI ≥30 kg/m², HbA1c ≥8%). Each threshold is treated as a binary indicator in the propensity score model. Patients without available BMI or HbA1c values are not excluded prior to matching; rather, they are classified as not meeting the specified threshold and are retained in the analysis. As a result, a complete-case analysis was not performed, and no patients were excluded solely due to missing laboratory data. The TriNetX platform does not support multiple imputation or alternative modeling strategies for missing laboratory values. Although matching on continuous laboratory measures would be preferable, incorporating BMI and HbA1c in binary form nonetheless allows partial adjustment for these clinically relevant variables and helps reduce confounding related to obesity and glycemic control.

Comment 5: Patients selected for GLP-1 RA therapy are likely to differ systematically from non-users in ways not fully captured by your covariates (e.g., care by endocrinologists, frequency of visits, socioeconomic status, lifestyle factors, detailed alcohol/tobacco exposure, CP severity, imaging intensity). While some of this is acknowledged in the Limitations, I would encourage you to expand the discussion of channeling bias, particularly how closer follow-up and more imaging in GLP-1 users might be expected to increase cancer detection rather than reduce it (which would, if anything, bias toward the null or against your observed direction). Also, consider an active comparator sensitivity analysis (e.g., GLP-1 RA vs another second-line diabetes agent such as SGLT2 inhibitor or DPP-4 inhibitor) if TriNetX allows, which would partially mitigate confounding by indication.

Response 5: Through subsequent analyses, we found that follow up time was actually shorter in the group that was in receipt of GLP-1 RA’s, thus channeling bias would not be applicable in our study. However, we do agree that those receiving GLP-1 RA’s vs those who are not are likely to differ systematically and these factors are included in our limitations.

Comment 6: The absolute number of pancreatic cancer cases in the GLP-1 groups is small (e.g., 20 events vs 50 in the matched CP cohort; 19 vs 37 in the matched CP+T2DM cohort). Please report exact event counts clearly in the text (they are visible in Table 2, but emphasizing this in the Results would help contextualize the ORs) and provide absolute risk differences and, if feasible, numbers needed to treat (NNT) with appropriate confidence intervals. The Discussion and Conclusion should temper the language around a “protective effect,” emphasizing that this is an association observed in an observational study with relatively few events, rather than definitive evidence of causality.

Response 6: Thank you for this important suggestion. To improve transparency and contextualize the effect estimates, we have now explicitly reported the absolute number of pancreatic cancer events for each cohort and absolute risk difference with 95% CI in the Results section, in addition to including this information in Figure 2. Given the relatively small number of events in the GLP-1–exposed groups, we agree that results should be interpreted cautiously. Accordingly, we have revised the Discussion and Conclusion to emphasize that the findings represent an association observed in an observational study, rather than evidence of a causal protective effect.

Comment 7: The Methods state that patients were excluded if they had prior GLP-1 RA exposure, prior pancreatic cancer, Whipple procedure, or bariatric surgery, and that at least two CP codes and one follow-up visit were required. Please clarify whether “at least two CP codes” had to occur prior to the index date and within a specified time window, and how recurrent or acute pancreatitis episodes were distinguished from chronic pancreatitis (K86.1 codes can be variably used). Also consider whether any minimum follow-up duration beyond the 6-month lag was enforced symmetrically in both groups. It would be helpful if Figure 1 could include the numbers and reasons for exclusions (prior pancreatic cancer, surgeries, missing key data) to increase transparency.

Response 7: We have clarified in the Methods section how and when two instances of K86.1 codes were used to ensure adequate follow-up and improve diagnostic accuracy. Unfortunately, it is not possible to reliably distinguish between acute and chronic pancreatitis solely based on K86.1 coding, as its use varies among healthcare providers. However, by requiring all patients to have at least two instances of this code, we are confident that our study population predominantly reflects patients with chronic pancreatitis rather than isolated acute episodes. Additionally, we have updated Figure 1 to provide detailed numbers and reasons for exclusions to increase transparency.

We have also provided more specific information as it relates to the inclusion/exclusion criteria.

Comment 8: You mention that large trials and observational studies have not consistently shown an association between GLP-1 RAs and pancreatic cancer. It would strengthen the paper to more systematically summarize the existing data specific to pancreatic cancer, distinguishing between safety signals (pancreatitis, pancreatic cancer) and emerging evidence of potential benefit. Please explicitly discuss how your findings in a high-risk CP/CP+T2DM population compare with findings in general T2DM populations (e.g., are your effect sizes larger, and if so, why might that be?).

Response 8: We now systematically summarize the existing data and evidence of potential benefit of GLP-1 RA’s specific to pancreatitis and pancreatic cancer. We also now discuss how our findings compare to existing literature in the general high-risk populations of T2DM and CP, individually.

Comment 9: In Table 1 and Table 3, the section header “GLP-1 Inhibitors” is misleading; these are GLP-1 receptor agonists, not inhibitors. Please correct the terminology. In Table 2, the wording “No GLP-1 vs. +GLP1” is a bit confusing; consider separate rows “GLP-1 RA users” and “Non-users” with explicit labels.

Response 9: The headers in the tables are now fixed and table 2 has been revised to improve readability.

Comment 10: Standardize p-value reporting (e.g., use “p < 0.001” rather than “p=0.0002” for very small values, or follow the journal’s preferred convention). Ensure consistent decimal places for ORs and CIs across text and tables.

 Response 10: These errors are now fixed.

Comment 11: Figure 1 (flowchart) is useful; consider slightly enlarging the font or simplifying the text boxes to improve readability. Indicating the numbers excluded at each step would improve transparency. You might consider adding a forest plot showing ORs (or HRs if time-to-event models are implemented) for CP overall and CP+T2DM, which would make the main results more visually accessible.

Response 11: The font is slightly bigger, as much as possible. Forest plots for HR’s are now implemented and demonstrated in the results and figures.

Comment 12: You already note several limitations; after addressing the major points above, please ensure that immortal time bias / time-alignment issues are explicitly mentioned, potential misclassification of CP, T2DM, and pancreatic cancer diagnoses from administrative codes is acknowledged, and the absence of information on GLP-1 RA dose, duration, and adherence is clearly stated as a key limitation.

Response 12: Thanks for the suggestions, these points are now addressed in the limitations section of our discussion.

Round 2

Reviewer 2 Report

Comments and Suggestions for Authors

-

Reviewer 4 Report

Comments and Suggestions for Authors

The authors have revised the manuscript in consideration of our suggestions. I have no further criticisms or suggestions.